Genome-wide identification and expression analysis of the VQ gene family in soybean (Glycine max)

Wang Yongbin 1 2
Jiang Zhenfeng 1
Li Zhenxiang 3
Zhao Yuanling 2
Tan Weiwei 2
Liu Zhaojun 2
Cui Shaobin 4
Yu Xiaoguang 4
Ma Jun 4
Wang Guangjin gjw1962@yeah.net 5
Li Wenbin wenbinli@neau.edu.cn wenbinli@yahoo.com 1
1 Key Laboratory of Soybean Biology in Chinese Ministry of Education, Key Laboratory of Soybean Biology and Breeding/Genetics of Chinese Agriculture Ministry, Northeast Agricultural University , Harbin , Heilongjiang , China
2 Biotechnology Research Institute, Heilongjiang Academy of Agricultural Sciences , Harbin , Heilongjiang , China
3 Harbin Normal University , Harbin , Heilongjiang , China
4 Heilongjiang Academy of Agricultural Sciences , Harbin , Heilongjiang , China
5 Soybean Research Institute, Heilongjiang Academy of Agricultural Sciences , Harbin , Heilongjiang , China
Uversky Vladimir
Electronic publication date: 2019 Aug 21
Publication date: 2019
Volume: 7
Electronic Location ID: e7509
Received 2019 Mar 18; Accepted 2019 Jul 17
Copyright: ©2019 Wang et al.
Copyright year: 2019
Copyright holder: Wang et al.
License: This is an open access article distributed under the terms of the Creative Commons Attribution License, which permits unrestricted use, distribution, reproduction and adaptation in any medium and for any purpose provided that it is properly attributed. For attribution, the original author(s), title, publication source (PeerJ) and either DOI or URL of the article must be cited.
License URL: https://creativecommons.org/licenses/by/4.0/

Keywords: VQ gene family, Glycine max, Gene expression, Phylogenetic analysis, Bioinformatics

Funding: National Transgenic Major Program of China 2016ZX08004001-006 National Key R&D Program for Crop Breeding 2016YFD0102105 Science and Technology Program for Innovation Talents of Harbin 2014RFQYJ016 2014RFXYJ011 Project of Heilongjiang Academy of Agricultural Sciences (HAAS) 2017BZ12 This work was supported by the National Transgenic Major Program of China (No. 2016ZX08004001-006), the National Key R&D Program for Crop Breeding (No. 2016YFD0102105), the Science and Technology Program for Innovation Talents of Harbin (No. 2014RFQYJ016, No. 2014RFXYJ011) and the Project of Heilongjiang Academy of Agricultural Sciences (HAAS) (No. 2017BZ12). The funders had no role in study design, data collection and analysis, decision to publish, or preparation of the manuscript.

==============================
Background

VQ proteins, the plant-specific transcription factors, are involved in plant development and multiple stresses; however, only few articles systematic reported the VQ genes in soybean.

Methods

In total, we identified 75 GmVQ genes, which were classified into 7 groups (I-VII). Conserved domain analysis indicated that VQ gene family members all contain the VQ domains. VQ genes from the same evolutionary branches of soybean shared similar motifs and structures. Promoter analysis revealed that cis-elements related to stress responses, phytohormone responses and controlling physical as well as reproductive growth. Based on the RNA-seq and qRT-PCR analysis, GmVQ genes were showed expressing in nine tissues, suggesting their putative function in many aspects of plant growth and development as well as response to stress in Glycine max.

Results

This study aims to understand the roles of VQ genes in various development processes and their expression patterns in responses to stimuli. Our results provide basic information in identification and classification of GmVQ genes. Further experimental analysis will allows us to know the functions of GmVQs participation in plant growth and stress responses.

Introduction

VQ genes are plant specific genes, which involved in plant development and multiple stress responses (Cheng et al., 2012). A conserved amino acid region has been identified within them, which composed of approximately 50–60 amino acids with a highly conserved the FxxhVQxhTG motif (Jing & Lin, 2015). The VQ domain possesses multiple biological functions in VQ proteins, such as the mutant strain of AtVQ14 (changes from IVQQ to EDLE) in the VQ domain result in producing small seeds, nevertheless the mutations in other locations does not have this characteristic (Wang et al., 2010). Furthermore, studies have reported that VQ genes are different in plants and do not have any intron in higher plants, whereas most VQ genes contain one or more introns in moss (Li et al., 2014; Jiang, Sevugan & Ramachandran, 2018; Dong et al., 2018).VQ proteins can interact with the WRKY proteins, for example, SIB1 and SIB2 are also VQ proteins, they were interacted with WRKY33 by recognizing the WRKY domain in C-terminal to activating the defense of plants (Lai et al., 2011).

VQ proteins were reported in dicotyledon such as Arabidopsis thaliana (Cheng et al., 2012), Vitis vinifera (Wang et al., 2015), Camellia sinensis (Guo et al., 2018), and monocotyledon such as Oryza sativa (Kim et al., 2013a; Kim et al., 2013b), Zea mays (Song et al., 2016). VQ proteins perform a variety of functions in plant development. For example, IKU1 (AT2G35230) is one of the VQ protein, it involved in regulating endosperm development and affect the seed formation during plant growth (Garcia & Berger, 2003). Under the far-red and low intensity of white light conditions, over expression of AtVQ29 can reduces the hypocotyl growth and it has higher expression in stem cells (Perruc et al., 1999). Furthermore, VQ genes regulate varying functions under abiotic and biotic stresses. AtCaMBP25 (also named AtVQ15) overexpression in transgenic plants had highly sensitive to osmotic stress in germination and early growth of seeds (Perruc et al., 1999). AtVQ9 alleviated the activity of WRKY8 under salt stress (Hu et al., 2013). The transcript levels of AtVQ23 and AtVQ16 are strongly induced by Botrytis cinerea infection and SA stress (Lai et al., 2011).

Glycine max is an important economic crop, widely cultivated in a number of countries. They are often subjected to abiotic stresses during the growth process, such as drought, high salinity, and other abiotic stresses were severely influenced on soybean production (Liu & Li, 2010). Therefore, identification of resistance genes has great significance for improving the yield and quality of soybean through molecular breeding. In this study, we identified 75 VQ genes of the soybean genome, and analyzed their phylogenetic, evolutionary motif, structure, promoter, and expression pattern. In addition, we analyzed the GmVQs’s expression level in different multiple abiotic stresses. Our results provide a basic information on identification and classification of GmVQ genes, and further experimental analysis allows us to comprehend the functions of GmVQs participate in plant growth and stress responses.

Materials & Methods

Identification of VQ genes

The Hidden Markov Model (HMM) profiles of the VQ motif PF05678 were downloaded from the Pfam database (Punta et al., 2012). HMM searched VQ motif (PF05678) from the G. max proteins database with the values (e-value) cut-off at 0.1 (Punta et al., 2012). The integrity of the VQ motif was determined using the online program SMART (http://smart.embl-heidelberg.de/) with an e-value < 0.1 (Letunic, Doerks & Bork, 2012). In addition, the three fields (length, molecular weight, and isoelectric point) of each VQ protein were predicted by the online ExPasy program (http://www.expasy.org/tools/) (Rueda et al., 2015).

Phylogenetic analysis

To investigate the phylogenetic relationship of the VQ gene families among A. thaliana, O. sativa, and G. max, AtVQ and OsVQ proteins were downloaded from phytozomes (http://www.phytozome.org) based on the previous studies (Cheng et al., 2012; Li et al., 2014; Goodstein et al., 2012). VQ proteins were aligned using the BioEdit program. A neighbor-joining (NJ) phylogenetic tree was constructed using these proteins through MEGA7.0 software (Tamura et al., 2011). Bootstrapping was performed with 1,000 replications. Genes were classified according to the distance homology with A. thaliana and O. sativa genes (Cheng et al., 2012; Li et al., 2014).

Sequence alignment, motif prediction and gene structure of GmVQ genes

Multiple alignments of the VQ full length proteins were conducted using Jalview software with default parameter settings. The online MEME analysis used to identify the unknown conserved motifs (http://meme.ebi.edu.au/meme/intro.html) using the following parameters: site distribution: zero or one occurrence (of a contributing motif site) per sequence, maximum number of motifs: 20, and optimum motif width ≥6 and ≤200 (Bailey et al., 2015). A gene structure displaying server program (http://gsds.cbi.pku.edu.cn/index.php) was used to show the structure of Glycine max VQ gene.

Gene duplication and collinearity analysis

The physical locations of the GmVQ genes on the soybean chromosomes were mapped by using MG2C website (http://mg2c.iask.in/mg2c_v2.0/). The analysis of synteny among the soybean genomes was conducted locally using a method similar to the one developed for the PGDD (http://chibba.agtec.uga.edu/duplication/) (Krzywinski et al., 2009). First, BLASTP, OrthoMCL software (http://orthomcl.org/orthomcl/about.do#release) and MCScanX software (Wang et al., 2012) were used to search for potential homologous gene pairs (E < 1 e−5, top five matches) across multiple genomes. Then, these homologous pairs were used as the input for the PGDD database (http://chibba.agtec.uga.edu/duplication/). Ideograms were created using Circos (Krzywinski et al., 2009).

Calculating Ka and Ks

The Ka and Ks were used to assess the selection history and divergence time of gene families (Li, Gojobori & Nei, 1981). The number of synonymous (Ks) and nonsynonymous (Ka) substitutions of duplicated VQ genes was computed by using the KaKs_Calculator 2.0 with the NG method (Xu et al., 2018). The divergence time (T) was calculated using the formula T = Ks∕(2 × 6.1 × 10−9) × 10−6 million years ago (MYA) (Kim et al., 2013a; Kim et al., 2013b).

VQ genes expression analysis of soybean

The expression data of VQ genes in different tissues, including seed, pod, SAM, stem, flower, leaf, root, root hair and nodule, is available in Phytozome V12.1 database (https://phytozome.jgi.doe.gov/pz/portal.html). The expression profile for VQ genes was utilized for generating the heatmap and k-means clustering using R 3.2.2 software (Gentleman et al., 2004).

Plant material and treatments

Glycine max (Williams 82) was used in this study. Seeds were planted in a 3:1 (w/w) mixture of soil and sand, germinated, and irrigated with half-strength Hoagland solution once every 2 days. The seedlings were grown in a night temperature of 20 °C and day temperature of 22 °C, relative humidity of 60 %, and a 16/8 h photoperiod (daytime: 05:00–21:00). After 4 weeks, the germinated seedlings were treated with 20% PEG6000 (drought), 250 mM NaCl solution (salt), 4 °C (cold), 100 µM abscisic acid (ABA), 100 µM salicylic acid (SA) solutions. Control and treated seedlings were harvested 1 h, 6 h, 12 h, and 24 h after treatment. All samples were frozen in liquid nitrogen and stored at −80 °C until use.

RNA extraction and Quantitative real-time PCR (qRT-PCR)

Total RNA was extracted from G. max using RNAiso Plus (TaKaRa, Tokyo, Japan) according to manufacturer’s instructions. The cDNA synthesis was carried out with approximately 2 µg RNA using PrimeScript RT reagent Kit with gDNA Eraser (TaKaRa, Tokyo, Japan). Quantitative Real-time PCR (qRT-PCR) was performed using SYBR Premix Ex Taq II (TaKaRa, Tokyo, Japan) on an ABI Prism 7000 sequence detection system (Applied Biosystems, USA) with the primers listed in Table S1. PCR amplification was performed in accordance with SYBR Premix Ex Taq (TaKaRa, Tokyo, Japan) response system. For each sample, three technical replicates were conducted to calculate the averaged Ct values. Relative expression was calculated by the 2−ΔΔCt method (Livak & Schmittgen, 2001). The actin and GAPDH genes were used as internal control.

Gene Ontology Enrichment

Once the sequences were obtained ran a BLASTX search against the UNIPROT database at a 1e-30 significance level. The matches were extracted and compared to the GO annotation generated against UNIPROT hits located at EBI. The GO annotation of the GmVQ genes by using WEGO 2.0 website (http://wego.genomics.org.cn/).

Analyzed the cis-elements of GmVQ promoters

The cis-elements of GmVQ promoters were analyzed to further understand the GmVQ gene family. We examined the sequences within 1,500 base pairs (bp) upstream of initiation codons (ATG) for promoter analysis and searched for these sequences in the soybean genome. The cis-elements in promoters were subsequently searched using the PlantCARE database (http://bioinformatics.psb.ugent.be/webtools/plantcare/html/).

Gene interaction network

Protein sequence of GmWRKY transcription factors were obtained from the genome database of soybean, also were mapped to the WRKY proteins of Arabidopsis by BLASTP tool in the TAIR database. Subsequently, the interaction between GmVQs and GmWRKYs were forecasted based on the PAIR website (https://rc.webmail.pair.com/), and their network was drawn in Cytoscape 3.6.1.

Results

Identification of GmVQs

Hidden Markov Model (HMM) of the VQ motif (PF05678) was used to search for putative VQs in soybean proteins database. A total of 75 VQs were identifiedand were named from GmVQ1 to GmVQ75 based on their physical locations on the chromosomes. This is different from the previous study, which 74 GmVQs were identified before the database updated (Wang et al., 2014; Zhou et al., 2016). ExPasy predicted that these 75 VQ proteins have different physical and chemical properties whose amino acid lengths ranged from 89 aa (GmVQ37) to 486 aa (GmVQ18), with an average of 223 aa and most of them were less than 300 aa. The molecular weights of these 75 VQ proteins ranged from 10.03 kDa (GmVQ37) to 52.79 kDa (GmVQ18) and their isoelectric points ranged from 4.29 (GmVQ69) to 10.74 (GmVQ51) (Table 1).

Table 1 List of all GmVQ genes identified in the Glycine max genome.

Gene name	Gene locus	Chromosome location	Length (aa)	pI	Molecular weight (Da)	Family group	
GmVQ1	Glyma01G018700	chr1:1790049-1792039	318	10.66	34712.78	VII	
GmVQ2	Glyma01G096800	chr1:31515839-31517715	289	10.24	31636.98	VII	
GmVQ3	Glyma01G195300	chr1:52952165-52953181	154	9.48	16878.08	VII	
GmVQ4	Glyma02G208800	chr2:39393500-39394691	212	9.96	23346.3	VII	
GmVQ5	Glyma03G120700	chr3:33242128-33243660	233	7.79	24442.59	VI	
GmVQ6	Glyma03G127800	chr3:34231323-34232268	167	4.79	18699.76	I	
GmVQ7	Glyma03G204900	chr3:41299415-41300202	119	9.84	13358.41	II	
GmVQ8	Glyma03G249100	chr3:44529232-44529956	127	9.11	14693.84	I	
GmVQ9	Glyma04G099600	chr4:9115245-9116947	287	10.12	31433.71	VII	
GmVQ10	Glyma04G103200	chr4:9567274-9568529	205	9.14	22493.86	V	
GmVQ11	Glyma04G103300	chr4:9570059-9571000	313	6.64	34199.48	V	
GmVQ12	Glyma04G134200	chr4:19214276-19215243	127	6.7	14519.25	I	
GmVQ13	Glyma04G214700	chr4:48626650-48627708	212	5.9	22930.94	II	
GmVQ14	Glyma04G239400	chr4:50786868-50788346	240	8.96	26255.85	II	
GmVQ15	Glyma05G107500	chr5:28551166-28551996	186	8.48	20503.94	VII	
GmVQ16	Glyma05G133000	chr5:32592583-32593521	211	7.79	23513.49	V	
GmVQ17	Glyma05G140700	chr5:33359443-33360263	113	5.43	12313.85	III	
GmVQ18	Glyma05G179700	chr5:36744975-36747458	486	6.12	52793.72	V	
GmVQ19	Glyma05G190000	chr5:37570564-37571544	208	6.84	22399.46	II	
GmVQ20	Glyma05G198400	chr5:38262465-38265127	186	9.52	20618.64	IV	
GmVQ21	Glyma06G101400	chr6:8043143-8044571	295	10.28	32310.83	VII	
GmVQ22	Glyma06G104400	chr6:8309457-8311289	341	6.06	37163.76	V	
GmVQ23	Glyma06G104500	chr6:8314638-8315685	316	6.48	34579.04	V	
GmVQ24	Glyma06G124400	chr6:10128263-10129012	249	8.11	27136.69	II	
GmVQ25	Glyma06G151400	chr6:12350255-12351217	222	5.97	24195.35	II	
GmVQ26	Glyma06G240300	chr6:39620687-39622549	244	7.79	26839.89	IV	
GmVQ27	Glyma07G028700	chr7:2307932-2309158	193	9.98	20948.74	IV	
GmVQ28	Glyma07G092500	chr7:8632559-8633302	247	5.97	27087.07	II	
GmVQ29	Glyma07G198000	chr7:36647302-36650373	310	8.42	33741.87	IV	
GmVQ30	Glyma08G005700	chr8:456890-461071	174	9.1	19216.73	IV	
GmVQ31	Glyma08G041900	chr8:3320834-3322022	140	6.9	15589.54	II	
GmVQ32	Glyma08G087400	chr8:6616587-6617687	221	6.91	24350.54	V	
GmVQ33	Glyma08G096000	chr8:7331414-7331749	111	6.26	12060.59	III	
GmVQ34	Glyma08G137300	chr8:10495440-10498069	472	6.33	51419.34	V	
GmVQ35	Glyma08G147600	chr8:11258747-11259761	198	6.51	21201.12	II	
GmVQ36	Glyma08G157900	chr8:12235183-12236400	141	5.63	15964.98	III	
GmVQ37	Glyma08G176500	chr8:14151104-14151373	89	7.89	10029.37	III	
GmVQ38	Glyma08G214100	chr8:17287863-17288952	194	9.69	21099.11	IV	
GmVQ39	Glyma08G272000	chr8:35627645-35629488	292	10.39	32000.2	VII	
GmVQ40	Glyma08G272100	chr8:35632723-35638206	361	9.8	39876.98	VII	
GmVQ41	Glyma08G272200	chr8:35665488-35667249	299	10.24	32844.1	VII	
GmVQ42	Glyma08G308400	chr8:42711855-42712403	182	4.3	20461.87	VII	
GmVQ43	Glyma09G051900	chr9:4508892-4509626	244	6.48	27252.83	V	
GmVQ44	Glyma09G111800	chr9:22128197-22129301	203	7.11	22686.53	III	
GmVQ45	Glyma09G183700	chr9:40881519-40882250	243	6.13	26618.56	II	
GmVQ46	Glyma10G273300	chr10:49575568-49576678	191	7.83	20981.58	II	
GmVQ47	Glyma11G046400	chr11:3468797-3469599	155	9.16	16952.31	VII	
GmVQ48	Glyma11G239600	chr11:33399330-33401730	439	7.02	47710.64	V	
GmVQ49	Glyma12G153600	chr12:23455875-23457485	248	7.02	27499.55	IV	
GmVQ50	Glyma12G225200	chr12:38479959-38482769	246	7.17	27184.17	IV	
GmVQ51	Glyma13G005100	chr13:1422443-1424046	224	10.74	23979.57	V	
GmVQ52	Glyma13G039800	chr13:12310527-12311616	240	10.11	25791.06	II	
GmVQ53	Glyma13G178500	chr13:29211898-29216138	281	9.8	30474.04	IV	
GmVQ54	Glyma13G193800	chr13:30709903-30710253	116	5.14	13478	I	
GmVQ55	Glyma13G218400	chr13:33181476-33181778	100	9.05	11045.26	III	
GmVQ56	Glyma13G238100	chr13:34835450-34840141	260	9.54	27989.72	IV	
GmVQ57	Glyma13G276100	chr13:37756820-37759545	249	7.91	27358.41	IV	
GmVQ58	Glyma14G002800	chr14:293552-294736	161	9.68	17490.39	V	
GmVQ59	Glyma14G124800	chr14:19432507-19433220	237	8.85	25366.52	II	
GmVQ60	Glyma14G172200	chr14:42617341-42619795	429	6.59	46407.55	V	
GmVQ61	Glyma15G075200	chr15:5769826-5772617	199	9.77	21595.31	IV	
GmVQ62	Glyma15G158200	chr15:13251793-13252987	252	7.16	27774.66	V	
GmVQ63	Glyma15G232200	chr15:43662201-43662912	122	6.65	14310.95	I	
GmVQ64	Glyma15G249800	chr15:47637825-47638440	89	7.89	10074.51	III	
GmVQ65	Glyma15G268300	chr15:50482677-50484349	158	7.74	17892.46	III	
GmVQ66	Glyma17G159600	chr17:13790434-13791675	190	9.3	21149.77	VII	
GmVQ67	Glyma17G182600	chr17:22616386-22616835	149	9.27	16873.05	V	
GmVQ68	Glyma18G017800	chr18:1285879-1287658	454	6.45	48264.96	V	
GmVQ69	Glyma18G108600	chr18:12426789-12427328	179	4.29	20004.4	VII	
GmVQ70	Glyma19G125300	chr19:38346007-38347425	232	9.64	24435.6	VI	
GmVQ71	Glyma19G130400	chr19:39031134-39032107	168	5.16	18878.02	I	
GmVQ72	Glyma19G202300	chr19:45923283-45923992	124	9.7	13533.53	II	
GmVQ73	Glyma19G246700	chr19:49331581-49332332	102	9.19	11775.23	I	
GmVQ74	Glyma20G064500	chr20:21930212-21930913	233	10.51	25070.57	V	
GmVQ75	Glyma20G116600	chr20:35927408-35928282	157	5.9	17333.32	II	

Phylogenetic analysis and multiple alignment of the VQ genes

To explore the phylogenetic relationships among the VQ genes of soybean, A. thaliana and O. sativa, a NJ phylogenetic tree was constructed (Fig. 1). We found that soybean and A. thaliana have a closer relationship than rice. Based on their relationship with AtVQs and OsVQs and the characteristics of GmVQs’ core domain, they were divided into 7 groups, designated Group I-VII (Figs. 1 and 2). For the 75 GmVQ proteins, Group VI contains two VQ proteins; Group V has the biggest amount, with 17 VQ proteins. Groups I, II, III, IV, VII contain 7, 15, 8, 12, 14 members respectively. At the same time, we found 5 types of VQ specificity domain: FxxxVQxLTG (54/75), FxxxVQxFTG (16/75), FxxxVQxVTG (2/75), FxxxVQxLTR (1/75), FxxxVQxLTS (1/75), besides, there is also a GmVQ protein (GmVQ10) has partial domain deletion (Fig. 2). Different types of VQ domains indicate that they might have different biological functions.

Figure 1 Phylogenetic tree analysis of the VQ genes in Glycine max, Arabidopsis thaliana and Oryza sativa.

The phylogenetic tree was constructed using MEGA 7.0 by the neighbor-joining method. The Bootstrap value was 1,000 replicates. The three plant-specific clusters were designated as group I-VII and indicated in a specific color.

Figure 2 Multiple sequence alignment, gene structure and multiple motifs of soybean.

Alignment of VQ domain of 75 VQ proteins in soybean. Amino acids that are conserved throughout are shaded in different colors. The genes in different groups are in different colors.

Conserved motifs and gene structures of the VQ gene family

We predicted that the 75 GmVQs contained 20 conserved motifs, with the motif length ranged from 11 aa to 50 aa (Fig. S1). Every GmVQ member contains 1-7 conserved motifs (Fig. 3B). All of the proteins, excepted GmVQ22, show motif 1 which contains a specialty VQ domain. Additionally, an unrooted phylogenetic tree was constructed with VQ protein sequences, suggested that the motifs organization of VQ genes were consistent with the phylogenetic tree (Fig. 3A). Group V contains motif 4, Group IV contains motif 2. We found that most groups possess more than two motifs, suggested that every group might have special functions with a highly conserved amino acid residue. Through the VQ gene structures analysis, half of the group VI has introns; genes in group V have longer coding regions, while genes of group I have shorter coding regions than other groups (Fig. 3C). Interestingly, 78.67% (59/75) of GmVQ genes are intronless genes. It is speculated that a large number of introns might be lost in VQ genes during evolution. The phylogenetic tree shows that genes from same branches have similar gene structures, while those from different branches have different gene structures (Fig. 3A).

Figure 3 Phylogenetic tree, conserved motifs and gene structure in GmVQs.

(A) Phylogenetic relationships (B) Conserved motifs of the GmVQs. Each motif is represented by a number in colored box. (C) Exon/intron structures of GmVQ genes.

Chromosome location and gene duplication

We drew a chromosomal location map of GmVQ genes on each chromosome (Fig. 4). GmVQs are distributed on all soybean chromosomes, except chromosome 16, and were densely distributed on chromosome 8 and chromosome 13, containing 13 and 7 members, respectively (Fig. 4). Most of GmVQs are distributed on the two ends of chromosomes.

Figure 4 Chromosome location and duplication events analysis in Glycine max.

Segmental or tandem duplicate in many gene families are the main expanding way in plants. To better study the evolution of GmVQ genes, we further explored gene duplication events using the MCScanX software. We found that 52 pairs of genes originated from segmental duplication, and 4 pairs of genes involved in tandem duplication events (Table S2).

Evolution and divergence of the VQ gene family in soybean and Arabidopsis

With the OrthoMCL software, we found 56 paralogous pairs in soybean, 37 orthologous pairs between soybean and Arabidopsis. Some VQ genes have never had any homology genes. All the paralogous and orthologous pairs are listed in Table 2. At the same time, we found that two or more GmVQ genes match to one AtVQ gene, implying that they might promote the expansion of the VQ gene family during evolution. We calculated Ka/Ks ratios of 55 paralogous pairs in soybean (Table 3). Most Ka/Ks ratios are <1, however, the GmVQ54/GmVQ63 and GmVQ65/GmVQ36 pairs are <1. In addition, the genetic differentiation of the 55 gene pairs occurred between 5 and 30 MYA.

Table 2 Paralogous (Gm-Gm) and orthologous (Gm-At) gene pairs.

Gm-Gm	Gm-Gm	Gm-At	
GmVQ3/GmVQ47	GmVQ24/GmVQ59	GmVQ37/AtVQ1	
GmVQ5/GmVQ70	GmVQ27/GmVQ38	GmVQ64/AtVQ1	
GmVQ6/GmVQ71	GmVQ28/GmVQ45	GmVQ14/AtVQ3	
GmVQ7/GmVQ72	GmVQ29/GmVQ53	GmVQ24/AtVQ3	
GmVQ8/GmVQ73	GmVQ29/GmVQ61	GmVQ52/AtVQ3	
GmVQ9/GmVQ21	GmVQ29/GmVQ56	GmVQ59/AtVQ3	
GmVQ10/GmVQ11	GmVQ34/GmVQ68	GmVQ29/AtVQ5	
GmVQ10/GmVQ22	GmVQ34/GmVQ48	GmVQ53/AtVQ5	
GmVQ10/GmVQ23	GmVQ37/GmVQ64	GmVQ61/AtVQ5	
GmVQ10/GmVQ67	GmVQ39/GmVQ40	GmVQ56/AtVQ5	
GmVQ11/GmVQ22	GmVQ39/GmVQ41	GmVQ46/AtVQ8	
GmVQ11/GmVQ23	GmVQ39/GmVQ2	GmVQ75/AtVQ8	
GmVQ11/GmVQ67	GmVQ40/GmVQ41	GmVQ9/AtVQ9	
GmVQ13/GmVQ25	GmVQ40/GmVQ2	GmVQ21/AtVQ9	
GmVQ14/GmVQ24	GmVQ41/GmVQ2	GmVQ37/AtVQ10	
GmVQ14/GmVQ52	GmVQ42/GmVQ69	GmVQ64/AtVQ10	
GmVQ14/GmVQ59	GmVQ43/GmVQ62	GmVQ27/AtVQ11	
GmVQ15/GmVQ66	GmVQ46/GmVQ75	GmVQ38/AtVQ11	
GmVQ16/GmVQ32	GmVQ49/GmVQ26	GmVQ1/AtVQ14	
GmVQ18/GmVQ34	GmVQ50/GmVQ57	GmVQ5/AtVQ15	
GmVQ18/GmVQ68	GmVQ51/GmVQ74	GmVQ70/AtVQ15	
GmVQ18/GmVQ48	GmVQ52/GmVQ59	GmVQ44/AtVQ17	
GmVQ19/GmVQ35	GmVQ53/GmVQ61	GmVQ50/AtVQ19	
GmVQ20/GmVQ30	GmVQ53/GmVQ56	GmVQ57/AtVQ19	
GmVQ22/GmVQ23	GmVQ54/GmVQ63	GmVQ28/AtVQ20	
GmVQ22/GmVQ67	GmVQ61/GmVQ56	GmVQ45/AtVQ20	
GmVQ23/GmVQ67	GmVQ65/GmVQ36	GmVQ19/AtVQ21	
GmVQ24/GmVQ52	GmVQ68/GmVQ48	GmVQ35/AtVQ21	
		GmVQ5/AtVQ24	
		GmVQ70/AtVQ24	
		GmVQ44/AtVQ25	
		GmVQ20/AtVQ31	
		GmVQ30/AtVQ31	
		GmVQ18/AtVQ34	
		GmVQ34/AtVQ34	
		GmVQ68/AtVQ34	
		GmVQ48/AtVQ34	

Table 3 Ka, Ks and Ka/Ks values calculated for paralogous VQ gene pairs.

Gene 1	Gene 2	Ka	Ks	Ka/Ks ratio	
GmVQ10	GmVQ11	0.002146692	0.013590406	0.15795643	
GmVQ39	GmVQ40	0.014329244	0.032130119	0.445975451	
GmVQ54	GmVQ63	0.070550548	0.066958938	1.053638991	
GmVQ65	GmVQ36	0.129398947	0.090241867	1.433912564	
GmVQ39	GmVQ41	0.015091052	0.092583143	0.163	
GmVQ37	GmVQ64	0.033904078	0.095366382	0.355513935	
GmVQ40	GmVQ41	0.02957328	0.096785754	0.305554062	
GmVQ42	GmVQ69	0.039057024	0.110714815	0.352771436	
GmVQ16	GmVQ32	0.056833182	0.117015316	0.485690114	
GmVQ40	GmVQ2	0.056768509	0.119061292	0.476800713	
GmVQ50	GmVQ57	0.012662345	0.127110953	0.099616473	
GmVQ49	GmVQ26	0.034291687	0.128532842	0.26679319	
GmVQ43	GmVQ62	0.057675771	0.129647567	0.444865819	
GmVQ19	GmVQ35	0.060650867	0.132943691	0.456214704	
GmVQ20	GmVQ30	0.047369299	0.134768712	0.351485879	
GmVQ7	GmVQ72	0.073853375	0.1352768	0.545942653	
GmVQ39	GmVQ2	0.050072412	0.136741168	0.366183886	
GmVQ18	GmVQ34	0.048240383	0.141323284	0.341347734	
GmVQ68	GmVQ48	0.072358564	0.146626964	0.493487433	
GmVQ3	GmVQ47	0.082715722	0.153202253	0.539911912	
GmVQ5	GmVQ70	0.053173881	0.15994977	0.33244112	
GmVQ46	GmVQ75	0.053993084	0.162730183	0.33179514	
GmVQ41	GmVQ2	0.047558234	0.164392307	0.289297198	
GmVQ10	GmVQ22	0.080947528	0.167701359	0.482688565	
GmVQ22	GmVQ23	0.12358775	0.173118789	0.713889871	
GmVQ51	GmVQ74	0.051867494	0.17525146	0.295960409	
GmVQ27	GmVQ38	0.068641212	0.187083237	0.366901992	
GmVQ8	GmVQ73	0.07651417	0.194357737	0.393676994	
GmVQ15	GmVQ66	0.064674193	0.201500105	0.32096357	
GmVQ9	GmVQ21	0.031810279	0.204712416	0.155390081	
GmVQ11	GmVQ22	0.068482759	0.20561255	0.33306702	
GmVQ28	GmVQ45	0.07428748	0.212707642	0.349246879	
GmVQ13	GmVQ25	0.0495425	0.21323249	0.232340297	
GmVQ61	GmVQ56	0.078572076	0.228081862	0.344490682	
GmVQ11	GmVQ67	0.187484416	0.242381253	0.773510383	
GmVQ14	GmVQ24	0.070535521	0.260458812	0.270812571	
GmVQ11	GmVQ23	0.143077885	0.278054006	0.514568687	
GmVQ10	GmVQ67	0.25396485	0.28526939	0.890263236	
GmVQ6	GmVQ71	0.098963625	0.304675089	0.324816923	
GmVQ22	GmVQ67	0.242648421	0.35967981	0.674623413	
GmVQ29	GmVQ53	0.1121903	0.365440906	0.306999841	
GmVQ10	GmVQ23	0.206060196	0.446610645	0.461386664	
GmVQ18	GmVQ48	0.431966031	0.855432366	0.50496807	
GmVQ18	GmVQ68	0.415845352	0.884745464	0.470016936	
GmVQ34	GmVQ68	0.40356234	0.937434864	0.430496406	
GmVQ34	GmVQ48	0.385776641	0.968352827	0.398384381	
GmVQ29	GmVQ61	0.320802002	0.987438675	0.324882963	
GmVQ24	GmVQ59	0.231784771	1.001714144	0.231388139	
GmVQ14	GmVQ59	0.271393455	1.079747859	0.251348917	
GmVQ23	GmVQ67	0.564635744	1.087320132	0.51929117	
GmVQ53	GmVQ61	0.281138217	1.090000891	0.257924759	
GmVQ29	GmVQ56	0.422850584	1.281581327	0.329944401	
GmVQ53	GmVQ56	0.280639233	1.337494009	0.209824665	
GmVQ14	GmVQ52	1.027115432	1.538918692	0.667426705	
GmVQ24	GmVQ52	1.074280372	1.754750076	0.612212751	

Expression analysis of GmVQ genes among various tissues

Sixty-seven GmVQ genes were investigated using available RNA-seq data from nine different tissues (seed, pod, SAM, stem, flower, leaf, root, nodule, and root hair) (Fig. 5). We found that the expression levels of the GmVQs varied significantly in different tissues.Most GmVQ genes were found expressed in more than one detected organ. As shown in Fig. 5, genes in group A are expressed in all analyzed tissues. The expression levels of group B in pod and stem tissues are higher. Genes in group C have specific expression in leaf and root.

Figure 5 Expression analysis of GmVQ genes in different tissues and different stages.

The clusters were designated as group A-C. Different colors in map represent gene transcript abundance values as shown in bar at top of figure.

Expression patterns of GmVQs under abiotic stress

We randomly selected 25 GmVQ genes from seven groups, and made sure their responses to the plant hormones-, cold-, salt-, and drought-stress (Figs. 6–10). Under ABA treatment, most genes were up-regulated whole treatment period and six genes (GmVQ6/8/31/33/59/71) were obviously down-regulated at some treatment time points (Fig. 6, Table S3). The expression levels of seven genes (GmVQ2/27/40/48/53/68/74) reached the peak at the 6 h treatment time point and four genes (GmVQ9/21/31/71) reached the lowest expression levels at the early treatment time points (0–1 h treatment). With SA treatment, the expression levels of most GmVQs were down-regulated throughout, while GmVQ7 was up-regulated at 1 h, 6 h and 12 h treatment time points (Fig. 7, Table S4). In addition, nine GmVQ genes (GmVQ5/6/8/23/31/68/70/71/74) were down-regulated under all abiotic stress.

Figure 6 qRT-PCR analysis reveals GmVQ genes under ABA treatment compared to the controls.

Stress treatments and time course are described in the Materials and Methods section. (A-Y) represent different genes which were used in qRT-PCR analysis. Asterisks on top of the bars indicating statistically significant differences between the stress and counterpart controls (*p < 0.05, **p < 0.01).

Figure 7 qRT-PCR analysis reveals GmVQ genes under SA treatment compared to the controls.

Stress treatments and time course are described in the Materials and Methods section. (A-Y) represent different genes which were used in qRT-PCR analysis.Asterisks on top of the bars indicating statistically significant differences between the stress and counterpart controls (*p < 0.05, **p < 0.01).

With cold treatment, the expression levels of fourteen GmVQ genes (GmVQ2/7/9/28/29/31/33/40/46/48/53/59/68/74) were up-regulated throughout (Fig. 8, Table S5), while the expression levels of three genes (GmVQ27/64/65) were down-regulated and then up-regulated during treatment. Under salt stress, the results were similar to that with cold stress treatment, most genes were up-regulated, eight genes (GmVQ9/23/27/33/65/68/70/71) were down-regulated throughout (Fig. 9, Table S6). On the contrary, under drought (PEG) stress, most genes were down-regulated, only eight genes (GmVQ2/6/7/8/21/29/33/48) were up-regulated during the treatment (Fig. 10, Table S7).

Figure 8 qRT-PCR analysis reveals GmVQ genes under cold treatment compared to the controls.

Stress treatments and time course are described in the Materials and Methods section. (A-Y) represent different genes which were used in qRT-PCR analysis. Asterisks on top of the bars indicating statistically significant differences between the stress and counterpart controls (*p < 0.05, **p < 0.01).

Figure 9 qRT-PCR analysis reveals GmVQ genes under NaCl treatment compared to the controls.

Stress treatments and time course are described in the Materials and Methods section. (A-Y) represent different genes which were used in qRT-PCR analysis. Asterisks on top of the bars indicating statistically significant differences between the stress and counterpart controls (*p < 0.05, **p < 0.01).

Figure 10 qRT-PCR analysis reveals GmVQ genes under drought treatment compared to the controls.

Stress treatments and time course are described in the Materials and Methods section. (A-Y) represent different genes which were used in qRT-PCR analysis. Asterisks on top of the bars indicating statistically significant differences between the stress and counterpart controls (*p < 0.05, **p < 0.01).

Cis-elements in GmVQ promoters

We found many hormone- and stress- related promoter’s cis-elements in GmVQ genes. Enhancer regions (CAAT-box) and core promoter element are around −30 bp of transcription start (TATA-box). Cis-acting regulatory element (A-box) are the common cis-acting elements in the promoter. Others cis-elements that were found in the 75 GmVQ s can be classified into three groups (Fig. 11). Twelve cis-elements involve in the hormone responsiveness; five cis-elements are stress-related elements: ARE/GC/LTR/MBS/TC; some GmVQ genes containe plant growth and development elements, such as CAT-box/circadian/GCN4/HD-Zip 1/MSA-like/RY-element. In addition, some GmVQ genes containe W-box motif, which is binding site for WRKY transcription factor.

Figure 11 Number of each cis-acting element in the promoter region (1.5 kb upstream of the translation start site) of GmVQ genes.

Gene Ontology Enrichment

To further understand the functions of the GmVQs, we performed GO annotation and GO enrichment analyses (Fig. S2 and Table S8). The GO terms included three categories, biological process (BP), molecular function (MF) and cellular component (CC). GO enrichment confirmed that these GmVQs were enriched in the biological process (GO:0008150), regulation of biological process (GO:0050789) and biological regulation (GO:0065007) terms of the BP category. Cellular component (GO:0005575), intracellular (GO:0005622) and cell (GO:0005623) were the most abundant functions in the CC category (Table S8). MF was enriched in molecular function (GO:0003674) and binding (GO:0005488). The GO ebrichment suggested that GmVQs were play curcial roles in regulated of biological process.

Gene interaction network analysis

Based on the PAIR tool, we found the functions and their interactions of the GmVQs and GmWRKYs. As shown in Fig. 12A, 3 GmWRKYs are supposed to interact with GmVQ proteins, included GmWRKY115, GmWRKY149 and GmWRKY156, all of them belong to WRKY’s IIc group. In the Fig. 12B, we found that GmWRKYs and AtWRKYs are quite similar in their core domains, indicated that they might have same functions, such as interacted with VQ proteins.

Figure 12 Interaction of GmVQ proteins with GmWRKY proteins.

(A) The prediction of interaction between GmVQ proteins and GmWRKY proteins by the PAIR website, and the interaction network was draw in Cytoscape 3.6.1. (B) Sequence analysis of the WRKY domains of GmWRKY proteins and AtWRKY proteins.

Discussion

VQ protein is a kind of specific protein that widely exists in plant, involved in plant growth and can response to different stresses (Petersen et al., 2010; Fiil & Petersen, 2011; Xie et al., 2010). Hence, we completed genome-wide analysis of soybean VQ proteins by bioinformatic analysis and qRT-PCR to understand their regulation when environmental changed. In the previous study, 74 GmVQ genes were identified (Wang et al., 2014; Zhou et al., 2016). After the database was updated, we identified and isolated 75 GmVQ genes in the soybean genome. Compared with previous study, the number of genes in chromosome 2, 4 and 17 show a big difference. Soybean contains more VQ genes than that of A. thaliana (34) (Cheng et al., 2012), Populus trichocarpa (51) (Chu et al., 2016) and O. sativa (42) (Kim et al., 2013a; Kim et al., 2013b). The reason is the whole genome duplication events (WGD). There are two rounds of genome duplication, occurred at around 59 and 13 million years ago, which caused 75% soybean genes duplicated (Jeremy et al., 2010).

Seventy-five VQ genes were identified in Glycine max’s genome, divided into seven groups based on their comprehensive phylogenetic tree among G. max, A. thaliana, and O. sativa. These proteins are in the shorter branches and with closer spacing, suggesting that they were highly conserved during the evolution. The more closer related genes within the same group shared more similar gene structures, either in their intron or in the exon patterns. Whereas, the variation in different groups suggested the functional diversity of the VQ genes (Jiang, Sevugan & Ramachandran, 2018). In addition, most GmVQ genes (59; 78.67%) were found intronless, and most GmVQ genes (64; 85.33%) encoded relatively small proteins with protein length less than 300 amino acid. This suggests that VQ gene families were intronless and they were highly conserved during evolution. At the same time, gene duplication can help plants to adapt to different environments during their development and growth (Huang et al., 2016; Storz, 2009). The main expansion of GmVQ gene family is segmental duplication (52; 92.9%), only 4 pairs of genes involved in tandem duplication events (4; 7.1%). A similar phenomenon was reported in the BrVQ gene family, which contains a high proportion of segmental duplication (71.9%) and low proportion of tandem duplication (28.1%) (Zhang et al., 2015).

Nonfunctionalization, subfunctionalization, and neofunctionalization generally take place after genome duplication, resulting in lose or fix of genes (He & Zhang, 2005; Sandve, Rohlfs & Hvidsten, 2018; Stark et al., 2017). Soybean has undergone the WGD and the whole genome triplication (WGT) compared to grapevine (Wang et al., 2017). As there are 18 VQ genes in grapevine genome, the predicted number of VQ genes in soybean should be more than 100 (Wang et al., 2015). However, in this study, we only found 75 VQ genes in the soybean genome, suggesting that there were gene loss events after genome duplication. In addition, the Ks value of each paralogous pairs was calculated to find gene duplication events, the most duplication events in GmVQ gene occurred between 5 and 30 MYA, consistent with the recent WGD in soybean (Wang et al., 2017; Jeremy et al., 2010). The Ka/Ks ratios in different gene pairs are different, but most gene pairs’ Ka/Ks ratios are less than one and only two gene pairs’ (GmVQ54-GmVQ63 and GmVQ65-GmVQ36) ratios are larger than 1, implying these gene pairs undergo different selection pressure. The above analysis indicated that purifying selection played a crucial role during the evolution, conserved VQ proteins evolved much slowly at the protein level.

Expression patterns of 67 GmVQ genes were performed to determine their tissue expression using RNA-seq data. The results showed that 24 genes were relatively highly expressed in nine tissues, indicated that they may relate to the growth and development of plants. Moreover, 76% (57/75) and 64% (48/75) of GmVQ genes’ expression levels were obviously increased in leaves and roots, respectively. More and more studies have shown that VQ proteins played a significant role in plants development. The study of A. thaliana mutants showed that AtVQ8 had a certain influence on chlorophyll formation and leaf growth and development (Cheng et al., 2012). In this study, GmVQ7 and GmVQ75 were in the same evolutionary branch with AtVQ8. Their high expression in leaves indicating they might have similar function as AtVQ8 (Cheng et al., 2012). These results will help us to study the further function of soybean’s VQ proteins.

Plants need to face various abiotic stresses during their growth in natural conditions, the most common of which are high salt, drought and cold (Kim et al., 2013a; Kim et al., 2013b; Wang et al., 2014). Except for regulation by environmental factors, VQ gene family is regulated by defense-related hormones, such as SA and ABA. In our study, we selected 25 GmVQs for qRT-PCR analysis under five different stresses (salt, drought, cold, SA and ABA stresses). In this study, most GmVQ genes were up-regulated with the SA treatment, the result is consistent with previous study that most AtVQ genes can response to pathogen or the SA treatment (Cheng et al., 2012). In addition, fifteen GmVQ genes (e.g., GmVQ2/21/29/46) were up-regulated under SA treatment, suggesting that they play a potential role in stress resitance. 56% GmVQ genes (14/25) were up-regulated, which is different with the up-regulation of VQ genes in rice that only three OsVQ genes were up-regulated more than two fold (Kim et al., 2013a; Kim et al., 2013b). Increasing evidence suggests that VQ genes are involved in various stress response. For example, 23% ZmVQ genes were upregulated, all the VvVQ genes were up-regulated by drought stress (Song et al., 2016; Wang et al., 2015). Consistently, 30% of GmVQ genes were up-regulated, GmVQ2/29/33 were highly expressed under drought stress. Nevertheless, AtVQ9 and AtVQ15 were reported can response to abiotic stress during high salinity treatment. The response of VQ genes to cold stress is similar to that of Chinese cabbage (Hu et al., 2013; Zhang et al., 2015; Cheng et al., 2012). In our study, GmVQ5/6/7/31/46/58/59 and GmVQ7/9/28 were activated the salt and cold stresses, respectively, because that their promoter region exists in specific stress cis-elements. Besides, homologous GmVQ genes possessed similar expression pattern but may exhibit opposite expression trend under stress, such as GmVQ9-GmVQ21 were up-regulated under SA treatment, but GmVQ9 was up-regulated and GmVQ29 was down-regulated during cold stress. These results suggest that GmVQ genes participate in response mechanism of abiotic stresses, their regulation mechanism is complex and diverse.

As auxiliary factor, VQ genes regulate transcription, can interact with many proteins to participate in regulating complex physiological and biochemical processes of plants, such as they can interact with WRKY transcription factors (Wang et al., 2015; Lei et al., 2017; Lai et al., 2011). Studies have shown that the responses under three different pathogens, VQ protein are interacted with WRKY protein in rice (Li et al., 2014). VQ proteins and WRKY proteins may form a protein complex to exercise function. We found some of the GmVQ genes interact with group I’s WRKY , most VQ genes interact with groups I and IIc’s VQ protein in various stresses in previous reports (Dong et al., 2018; Guo et al., 2018; Lei et al., 2017). The promoter analysis indicated that 23 of 75 GmVQ genes (30.67%) contained one or more W-box motif in their 1,500 bp promoter regions, W-box were present in 78% VvVQ genes, 91% ZmVQ genes contained one or more W-box motif (Song et al., 2016). In the promoters of GmWRKY genes, W-boxes could regulate GmWRKY members (Dong, Chen & Chen, 2003). It indicates that WRKY protein affect VQ genes expression and thus responses to environmental stimuli (Dong et al., 2018; Guo et al., 2018).

Conclusions

Seventy-five VQ genes were identified in the soybean genomes. All VQ genes fell into seven groups (I-VII). VQ genes from the same evolutionary branches of soybean shared similar motifs and structures. The selection pressure analysis showed that most of the paralogous pairs were under a strong purifying selection in the GmVQ genes. RNA-seq analysis revealed that the VQ genes had different expression patterns in different tissues, indicating that they play crucial roles in different tissue. Finally, qRT-PCR showed that the VQ gene family was responsive to biotic and abiotic stresses. Our results provide a theoretical basis for further study on the function of GmVQs.

Supplemental Information

Figure S1 Sequence logo of motifs in GmVQ genes

Click here for additional data file.

Figure S2 Gene ontology categories assigned of the GmVQ genes

Click here for additional data file.

Table S1 List of primers used in qRT-PCR

Click here for additional data file.

Table S2 List of VQ gene duplication events

Click here for additional data file.

Table S3 Raw data for the ABA stress

Click here for additional data file.

Table S4 Raw data for the SA stress

Click here for additional data file.

Table S5 Raw data for the salt stress

Click here for additional data file.

Table S6 Raw data for the drought stress

Click here for additional data file.

Table S7 Raw data for the cold stress

Click here for additional data file.

Table S8 GO terms of the GmVQ genes

Click here for additional data file.

The authors would like to thank the key laboratory of crop and livestock molecular breeding of Heilongjiang Province for providing plenty of helpful manpower and material support.

Additional Information and Declarations

Competing Interests

Author Contributions

Data Availability

The authors declare there are no competing interests.

Yongbin Wang performed the experiments, analyzed the data, contributed reagents/materials/analysis tools, prepared figures and/or tables, authored or reviewed drafts of the paper, approved the final draft.

Zhenfeng Jiang performed the experiments, analyzed the data, authored or reviewed drafts of the paper, approved the final draft.

Zhenxiang Li analyzed the data, contributed reagents/materials/analysis tools, prepared figures and/or tables, authored or reviewed drafts of the paper, approved the final draft.

Yuanling Zhao and Weiwei Tan performed the experiments, prepared figures and/or tables, approved the final draft.

Zhaojun Liu contributed reagents/materials/analysis tools, prepared figures and/or tables, approved the final draft.

Shaobin Cui, Xiaoguang Yu and Jun Ma contributed reagents/materials/analysis tools, approved the final draft.

Guangjin Wang and Wenbin Li conceived and designed the experiments, approved the final draft.

The following information was supplied regarding data availability:

The raw measurements are available in Fig. S1 and Table S1.

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
