# Peer review of "Genome-wide identification and expression analysis of the VQ gene family in soybean (Glycine max)"

_PeerJ, doi:10.7717/peerj.7509_

## Round 0.1 · original submission · Major Revisions

Please address all critiques of reviewers and revise manuscript accordingly.

Reviewer 1 ·

Basic reporting

The manuscript shows professional organization and sufficient reference and background. All the data were analyzed and interpreted appropriately. The figures and tables are shown with good quality.

Experimental design

Methods described in the manuscript is sufficient to follow. Research question well defined, relevant and meaningful.

Validity of the findings

The impact and novelty are suitable for the publication. Data was interpreted completely and supported the conclusion.

Additional comments

Overall, the manuscript contains sufficient data to support its conclusion. The organization of the full text is scientific and professional for publication. The authors need to consider polishing the language for a better understanding of international researchers.

Reviewer 2 ·

Basic reporting

This paper titled “Genome-wide identification and expression analysis of the VQ gene family in soybean (Glycine max)” reported the VQ genes in soybeans, which may inform further research. Generally, the paper is with scientific validity and adequate data on the topic. The reviewer has a few minor comments for the authors to consider.

1. Please proofread to avoid topo or grammar mistakes. For instance,
Line 52-54: "GmVQ genes were expressed in nine tissues suggested their putative function in many aspects of plant growth and development, and response to stresses in Glycine max" to "GmVQ genes that were expressed in nine tissues suggested their putative functions in many aspects of plant growth and development, and response to stresses in Glycine max"
Line 85-86: "cultivated in the A number of countries" to "cultivated in a number of countries"
2. Kindly expand the figure legends with essential information such as statistical significance and sample size if necessary.

Experimental design

no comment

Validity of the findings

no comment

·

Basic reporting

No comments

Experimental design

No comment

Validity of the findings

No comments

Additional comments

In this manuscript, the authors carried out the genome-wide identification of the VQ motif gene family in the soybean genome Glycine max. They identified 75 VQ genes. They then classified these members, analyse their cis-elements in their promoter regions, chromosome location and gene replication as well as protein interactions with WRKY proteins. They then mainly focused on expression analysis using publicly available expression data as well as their own qRT-PCR data. They tried to analyse expression patterns at different developmental stages and expression regulation under various abiotic stresses and hormone treatments. Generally, the authors carried out some bioinformation analysis by using publicly available genome annotation databases and expression databases. Their works may provide some useful information. However, VQ genes have been identified but the authors did not mention it. Some comments are listed as below.
Genome-wide identification and evolutionary as well as expression analysis of the VQ gene family have been carried out in many many plant species. In the soybean genome, at least two reports have been published on the genome-wide identification of the VQ genes: (i) Wang, X., Zhang, H., Sun, G., Jin, Y., and Qiu, L. 2014. Identification of active VQ motif-containing genes and the expression patterns under low nitrogen treatment in soybean. Gene 543(2):237-243; (ii) Zhou Y, Yang Y, Zhou X, Chi Y, Fan B, Chen Z. 2016. Structural and Functional Characterization of the VQ Protein Family and VQ Protein Variants from Soybean. Sci Rep. 2016 6: 34663. The reference Wang et al (2014) was listed but was wrongly cited. The authors did not mention that the VQ gene family members were identified and were reported in at least these two references.
By raw comparison between the reference by Zhou et al (2016) and this study, the former identified 74 VQ genes and this study identified 75 genes. In chr2, chr4 and chr17, Zhou et al (2016) identified 2, 5 and 1 VQs but this study identified 1, 6, and 2 VQs. The authors at least need to make a comparison among published VQs and their identified VQs. Acturally, the 75 VQs can be easily available in the Phytozome website (https://phytozome.jgi.doe.gov/pz/portal.html#!results?search=0&crown=1&star=1&method=4433&searchText=vq&offset=0), which need to be further verified.
The only valuable data should be the expression data under various abiotic stresses or hormone treatments by qRT-PCR, which may be useful for someone who are interested in these VQ genes for their functional characterization.
Wang et al (2015) was cited but no reference was listed.
Wang et al (2017) was cited but two references were listed.

Reviewer 4 ·

Basic reporting

See comments

Experimental design

See comments

Validity of the findings

See comments

Additional comments

In this work, the authors used bioinformatics tools to analyze the VQ proteins from Glycine max, and determined their expression level in different abiotic stress conditions.

The manuscript is difficult to follow as almost every other line in the paper has one or more English errors. Please thoroughly proof-read the manuscript and use unambiguous and professional English. For most part the methods are sufficiently detailed but at places require some clarification as noted in general comments below. Overall, the manuscript requires major revision and can only be considered again provided authors address my concerns listed below:
1) Introduction needs major improvement and should be rewritten as sufficient background is not provided to determine significance of the work. Previous literature characterizing the VQ protein from Glycine max has not been discussed (see for example: doi: 10.1038/srep34663). Authors should clarify how the present work adds to what is already known in the field.
2) On lines 55-56 and on line 91, “The present study provided basic information for further analysis of the biological …”. I find this statement vague. What do authors mean by basic information here?
3) On lines 96-99, “HMM searched VQ motif (PF05678) from the G. max protein database with values (e-value) cut-off at 1.0 (Punta et. al, 2012). The integrity of the VQ motif was determined using the online program SMART (…) with an e-value < 0.1”. It is not clear what authors want to say here. Can authors elaborate on why different e-values is being used.
4) On line 105, “… VQ protein sequences were downloaded from phytozomes….”. It is not clear why the sequences retrieved using Pfam, as described in previous section under “Identification of VQ gene information”, were not used for phylogenetic analysis.
5) On line 106, “…VQ genes were aligned …”. Since protein sequences were downloaded, I assume author aligned VQ proteins and not VQ genes.
6) On line 107, “… neighbor-joining (NJ) phylogenetic tree was constructed using the ….”. Was the tree constructed using the amino acid sequences downloaded from phytozomes?
7) On line 143, version number for the R software used should be provided.
8) On line 183, “We identified 75 VQ genes in soybean genome…”. Authors should expand on how these 75 genes were identified? Previous studies (see for example: doi: 10.1038/srep34663) have also reported 74 VQ proteins in G. max. Authors should comment on whether the sequences identified here are similar/different from those previously characterized.
9) On line 192, “…they were divided into 7 groups, designated …”. Authors should expand on how the 7 groups were identified.
10) On line 195, “…we found 5 types of VQ specificity domain (Fig. 2)”. From figure 2, it is not clear what 5 types of specificity domains author is talking about. They should elaborate more on what these domains are and their possible significance. Also, what do different colors mean in Figure 2? This should be added in the figure legend.
11) On lines 199-200, “….all the proteins excepted GmVQ22 shown motif 1, which is contained a specialty VQ domain.” Isn’t the presence of VQ motif a characteristic of VQ proteins? If so, why is this protein classified as VQ protein without this motif?
12) In figure 3A, why is group VII split into two branches? Shouldn’t these sequences then be classified as two separate groups based on the tree presented in this figure?
13) Authors should add to Figure 4 legends what magenta lines in the figure represent.
14) On line 219-220, “…we found that 31 pairs of genes originated from segmental duplication, and 4 pair of genes involved in tandem duplication events.” Authors should elaborate on how they arrived at this observation.
15) On lines 224-225, authors should describe how they found the paralogous and orthologous pairs.
16) On line 242, authors should describe how the 25 genes were selected for abiotioc stress response determination.
17) On line 274-275, “….our results proved GmWRKY is similar to AtWRKY and those proteins can interacted and VQ protein”. Authors should understand that they are not proving anything here. The interactions shown are merely predictions which are not experimentally verified. Also, authors should elaborate on why they think GmWRKY is similar to AtWRKY. Authors show sequence comparison in Figure 12, but do not discuss it in the manuscript.
18) On lines 286-288, “These proteins are highly conserved because of shorter branches and closer spacing, we assume that they have similar functions”.It is not clear what authors are trying to convey here.

---

## Round 0.2 · Minor Revisions

As you can see, both reviewers accepted your manuscript scientifically. However, one of the reviewers indicated that your manuscript requires careful proofreading, as it contains linguistic issues. Therefore, you need to make some steps to fix this problem (e.g., contact professional editors or ask a colleague, who is fluent English speaker to edit your manuscript).

[]

·

Basic reporting

Good reporting with clarity

Experimental design

Well defined

Validity of the findings

The findings are validated and novel

Additional comments

The authors have taken care of my comments and the manuscript has improved from previous version

Reviewer 4 ·

Basic reporting

NA

Experimental design

NA

Validity of the findings

NA

Additional comments

Authors have addressed all my concerns. However, the manuscript still lacks in professional English with several grammatical mistakes. I would recommend publishing the manuscript provided authors proofread and correct for mistakes in English.

---

## Round 0.3 · Minor Revisions

The manuscript was carefully edited and the linguistic issues were fixed.

However, a final check by Section Editor Gerard Lazo has resulted in the following comment/request:

“There are 75 new VQ genes identified which have been associated with different stress responses and tissue types. It is important that these new sequences be associated with the appropriate annotation terms so that they can be compared to other model systems. We request that terms be added relative to the appropriate gene ontology (GO) biological, functional, and cellular terms. Perhaps another supplementary file can be added, or a new column for the GO terms in the raw data tables would suffice. Journal manuscripts are often scanned by text-mining software that locates and extracts core data elements, like gene function. Adding standard ontology terms, such as the Gene Ontology (GO, geneontology.org) or others from the OBO foundry (obofoundry.org) can enhance the recognition of your contribution and description. This will also make human curation of literature easier and more accurate. None of this was currently visible."

Therefore, please address this final request before we can Accept the manuscript.

---

## Round 0.4 · Minor Revisions

Here are comments I received from Gerard Lazo, the Section Editor:

"I looked at the VQ sequences and was able to extract the genome regions from the Phytozome v12 genome for soybean. Once the sequences were obtained I ran a BLASTX search against the UNIPROT database at a 1e-30 significance level. The matches were extracted and compared to the GO annotation generated against UNIPROT hits located at EBI. Matches were generated for values in the cellular, functional, and molecular GO categories. These values may be significant and are based on the GO evidence codes classified as Inferred from Electronic Annotation (IEA), but could be curated to be a more specific code to elevate to other inference levels, such as Inferred from Sequence Alignment (ISS) based on the expertise of the authors. Rather than having nothing done for this request, it would be of value to have such terms incorporated for others interested to carry the work forward. There were over 350 terms created in doing such an exercise.

Attached are the UNIPROT GO annotations (molecular, cellular, biological) which best matched to the sequences contained in the extracted genome sequences from soybean (VQgenes.fasta). The BLAST report is attached as an m6-generated table from BLASTX+. Perhaps these values can be added to the table which had nothing entered. This could have very well been done by someone in your group. You will need to validate the matches and the terms.
The extracted sequences did however match regions which were aligned to VQ regions based on the coordinates provided in the table. Some data or the like should have been generated as supplemental files.

Other strategies would be to take note of the expression characteristics and try to apply them with respect to tissue, development, or function."

Dr Lazo has dedicated considerable time to this feedback and analysis. Please address these points and amend your manuscript accordingly.

By separate email, staff will forward you the files created by Dr Lazo.

---

## Round 0.5 · accepted · Accept

Thank you very much for adequately addressing the remaining critical points and for revising your manuscript accordingly.

#